# Short and Long-Term Surgical Outcomes of Laparoscopic Total Gastrectomy Compared with Open Total Gastrectomy in Gastric Cancer Patients

**DOI:** 10.3390/cancers15010076

**Published:** 2022-12-23

**Authors:** Sang Soo Eom, Sin Hye Park, Bang Wool Eom, Hong Man Yoon, Young-Woo Kim, Keun Won Ryu

**Affiliations:** Center for Gastric Cancer, National Cancer Center, Goyang 10408, Republic of Korea

**Keywords:** gastric cancer, laparoscopic gastrectomy, total gastrectomy

## Abstract

**Simple Summary:**

Laparoscopic total gastrectomy (LTG) remains controversial in terms of its short- and long-term surgical outcomes in comparison to open total gastrectomy (OTG). This study aimed to compare the outcomes of LTG with OTG. There was no significant difference in short-term outcomes between the two groups. Additionally, the 3-year disease-free survival and 5-year overall survival rates were not significantly different between the two groups. Therefore, LTG could be an alternative approach to OTG.

**Abstract:**

This study aimed to compare the efficacy of laparoscopic total gastrectomy (LTG) with that of open total gastrectomy (OTG) in terms of postoperative complications and long-term survival. We retrospectively reviewed the clinicopathological data of 560 patients, who underwent total gastrectomy between 2012 and 2016 at the National Cancer Center, Korea. Propensity-score matching (PSM) was performed to correct for discrepancies between the two groups. Matched variables included sex, age, body mass index, American Society of Anesthesiologists score, and pathological Tumor–Node–Metastasis stage. After PSM, 238 patients were included in this analysis. The rate of D2 lymph node dissection was significantly higher in the OTG group than in the LTG group. The estimated blood loss was significantly lower in the LTG group than in the OTG group. The overall complication rate was not significantly different between the two groups. There was no significant difference in the 3-year disease-free and 5-year overall survival rates between the two groups. LTG and OTG had comparable efficacies in gastric cancer patients regarding short- and long-term surgical outcomes. This study suggests that LTG could be an alternative approach to the OTG.

## 1. Introduction

Gastric cancer is the fifth most common cancer worldwide and the fourth most common cause of cancer-related death [1]. The incidence of upper stomach and esophagogastric junction (EGJ) cancer is increasing worldwide, particularly in the western world and with similar trends in eastern countries [2,3,4,5,6]. Therefore, the use of total gastrectomy (TG) for the treatment of proximally located gastric cancer is also increasing [7]. However, TG has been considered technically challenging in esophagojejunostomy, which can lead to mortality if anastomosis leakage occurs [8]. Particularly in advanced gastric cancer (AGC), TG requires more skills to achieve oncological safety due to difficult lymph node dissection (LND) near the mediastinum or spleen.

As surgical techniques evolve, minimal invasive surgery (MIS) has been expected to have the advantages of reducing postoperative complications and recovery time in various intra-abdominal diseases [9,10]. In several large phase III randomized trials, such benefits were also proven in gastric cancer surgery in terms of postoperative morbidity and mortality in distal subtotal gastrectomy, regardless of disease status [11,12]. Moreover, the long-term survival of laparoscopic distal gastrectomy (LDG) is similar to that of open distal gastrectomy (ODG) in both early gastric cancer (EGC) and operable AGC [13,14,15,16]. Based on the results of these studies, laparoscopic surgery is recommended as an alternative to open surgery in distal gastrectomy for operable gastric cancer [17,18]. However, the laparoscopic approach for TG has been controversial owing to technical difficulties, such as LND and reconstruction, compared to open surgery [19]. To date, no prospective randomized study has compared the efficacy of laparoscopic total gastrectomy (LTG) with that of open total gastrectomy (OTG). Most studies were non-randomized cohort studies or case-control studies. Therefore, the efficacy of LTG is questionable, and the recommendation of LTG in different guidelines remains a matter of controversy [7].

In this study, the efficacy of LTG was evaluated by comparing the short- and long-term surgical outcomes to those of OTG after propensity-score matching (PSM) in a large-volume center. Short-term postoperative complications and long-term survival were compared as parameters of LTG efficacy.

## 2. Materials and Methods

### 2.1. Patients

A total of 578 patients, who underwent TG for gastric adenocarcinoma between January 2012 and December 2016 at the National Cancer Center, Korea, were analyzed. Among them, 18 patients were excluded due to loss of follow-up. No patients had received neoadjuvant chemotherapy and upfront surgery. The Korean practice guidelines recommend D1 + LND in patients with clinical stage IA gastric cancer and D2 LND in patients with clinical stage IB, II, and III gastric cancer. Patients with pathologic stage II or III were indicated for adjuvant chemotherapy, except for those with old age or concerns regarding side effects of chemotherapy. In our center, patient treatments are based on this clinical practice guideline. Patients were divided into two groups according to the surgical method (OTG: *n* = 409, LTG: *n* = 151). Laparoscopic-assisted total gastrectomy (LATG) and total laparoscopic total gastrectomy (TLTG) were included in the LTG group. Cases that initially tried by laparoscopy surgery but were converted to open surgery for any reasons were included in OTG group. In LATG, an additional epigastric longitudinal incision was made in the anastomosis. A circular end-to-end anastomosis stapler was used for end-to-side anastomosis after inserting the envil into the esophageal stump. In TLTG, a linear stapler was used for side-to-side anastomosis, and the entry hole was closed by continuous suture or a linear stapler without additional incision [20]. This study was approved by the Institutional Review Board of the National Cancer Center (Approval Number: NCC 2022-0038). Because this was a retrospective study, the requirement for informed consent was waived.

### 2.2. Outcome Assessments

Perioperative outcomes were collected, including age, sex, body mass index (BMI), American Society of Anesthesiologists (ASA) score, pathologic Tumor–Node–Metastasis (TNM) stage, tumor size, adjuvant chemotherapy, LND level, tumor location, tumor histology, operating time, estimated blood loss (EBL), positive lymph node (LN), dissected LN, complications occurring within 30 days after surgery using the Clavien–Dindo classification system, proximal resection margin (PRM), distal resection margin (DRM), hospital day (HD). Survival outcomes were also analyzed, including 3-year disease-free survival (DFS) and 5-year overall survival (OS). Histological types were classified according to the 2010 World Health Organization classification. Staging was classified according to the 8th edition of the American Joint Committee on TNM Manual. DFS was defined as the time after treatment, during which no signs of cancer were detected. OS was defined as the percentage of patients in the treatment group who were alive after treatment initiation.

### 2.3. Statistical Analysis

All statistical analyses were performed using R software version 4.1.2 (R Core Team [20,21], R: Language and environment for statistical computing, R Foundation for Statistical Computing, Vienna, Austria). Continuous variables, presented as means and standard deviations or medians and percentiles, were compared using a t-test or Wilcoxon rank sum test. Categorical variables, presented as frequencies and percentages, were compared using the χ^2^ test and Fisher’s exact test. The Kaplan–Meier method was used to estimate the survival outcomes. Statistical significance was set at *p* < 0.05. PSM was performed to correct for discrepancies between the two groups. Matched variables included sex, age, BMI, ASA score, and pathological stage. After matching, the balance between the two groups was examined using the standard mean difference (SMD). If the SMD value was < 0.1, the matching was considered to be well balanced.

## 3. Results

### 3.1. Clinicopathological Characteristics

Clinicopathological characteristics of the patients were summarized in Table 1. Of the 560 patients, 409 were included in the OTG group and 151 were included in the LTG group. Before PSM, the LTG group had significantly more female patients (37.8% vs. 27.9%, *p* = 0.0243), higher tumor location (*p* = 0.0003), lower pathologic stage (*p* < 0.0001), smaller tumor size (*p* < 0.0001), less LND (*p* < 0.0001), and less adjuvant chemotherapy (*p* < 0.0001) than the OTG group. Age, BMI, ASA score, and histology were not significantly different between the two groups. After PSM, the LTG group included 119 patients and the OTG group included 119 patients. Stage I, II, and III patients were 93 (78.2%), 17 (14.3%), and 9 (7.6%), respectively in both groups. Even after PSM, the LTG group had a significantly higher tumor location (*p* = 0.0032) and lower LND (*p* = 0.0025) than the OTG group. There were no significant differences in other clinicopathological data between the two groups after PSM including pathologic stage and adjuvant chemotherapy.

### 3.2. Surgical Outcomes

Table 2 summarized the postoperative surgical results. Before PSM, there is significantly lower EBL (50 vs. 200 mL, *p* < 0.0001), less positive LN (0 vs. 1, *p* < 0.0001), less dissected LN (39 vs. 44, *p* = 0.0163), longer DRM (12.2 vs. 8.8 cm, *p* < 0.0001), and shorter HD (7 vs. 8, *p* < 0.0001) in the LTG group than in the OTG group. After PSM, the LTG group had significantly lower EBL (50 vs. 180 mL, *p* < 0.0001), shorter length of PRM (3.1 vs. 3.7 cm, *p* = 0.0491), and longer length of DRM (12.6 vs. 10 cm, *p* = 0.0034) than the OTG group. The operating time (205 vs. 200 min, *p* = 0.2440), hospital stay duration (7 vs. 8 days, *p* = 0.1396), positive LN (0 vs. 0, *p* = 0.5135), and dissected LN (38 vs. 37, *p* = 0.3058) were not significantly different between the two groups.

Table 3 showed postoperative surgical complications within 30 days from surgery after PSM. Complications were classified using the Clavien–Dindo classification system. The overall complication rates were not significantly different between the two groups (25.2% vs. 34.5%, *p* = 0.1191). However, there were fewer complications of Clavien–Dindo grade I in the LTG group than in the OTG group (3.4% vs. 11.0%, *p* = 0.0224). There were no significant differences in the rates of other complication grades, including severe complications (> Clavien–Dindo grade IIIa), between the two groups (14.3% vs. 17%, *p* = 0.5722).

### 3.3. Survival Analysis

There were no significant differences in the 3-year DFS and 5-year OS between the two groups after PSM. The 3-year DFS rates were comparable between the two groups (LTG vs. OTG: 88.69% vs. 85.56%, *p* = 0.4467). In addition, 5-year OS rates were similar between patients who underwent OTG and those who underwent LTG (LTG vs. OTG: 83.23% vs. 79.79%, *p* = 0.5258) (Figure 1).

## 4. Discussion

The present study compared postoperative complications and long-term survival of gastric cancer patients who underwent TG by open or laparoscopic surgery to evaluate surgical efficacy after PSM. There were no significant differences in postoperative complications, 3-year DFS, and 5-year OS between the two surgical methods. These findings suggested that LTG may be an alternative approach to OTG and LDG in gastric cancer.

Pang et al. [21] reported the safety of LTG in T4a gastric cancer patients in terms of long-term survival. Similarly, Gambhir et al. [22] compared OTG and LTG in stage I–III gastric cancer from a national database; however, most EGC patients underwent LTG, while AGC patients underwent OTG. Few studies have compared the long-term outcomes of LTG and OTG by equal matching at all stages. In the absence of prospective study outcomes, our study compared the short- and long-term outcomes after PSM in all patients to minimize bias in large-volume single centers. Therefore, well-balanced groups after PSM (data shown in Table 1, Table 2 and Table 3) were used for comparative analysis in our study.

Because OTG was considered a standard treatment for AGC during the early period in this study [23], OTG was performed more frequently than LTG in AGC. These two groups were compared in a balanced manner after successful matching. However, even after PSM, there was significantly less D2 LND in the LTG group than in the OTG group. We hypothesized that this finding might result from the difficulty of LND in LTG or the difference in clinical stages between the two groups, which might have occurred because 80% of the patients were in the stage I group after PSM.

In a Korean nationwide survey conducted on surgically treated gastric cancer patients in 2019, the frequency of the laparoscopic approach increased from 48.1% in 2014 to 64.9% in 2019, whereas the frequency of the open approach decreased from 49.8% to 27.6% in the same time period [3]. The current trend is shifting from open surgery to MIS because surgeons can expect various benefits from MIS. According to a subgroup analysis of a survey conducted in Korea, LDG frequency increased by 20.8% to 77.2% in 2019 in comparison to its frequency in 2014 (56.4%), and LTG frequency increased by 18.2% in to 44.3% in 2019 in comparison to its frequency in 2014 (26.1%) [3,24]. Along with LDG, the laparoscopic approach for both TG and LDG was attempted, but the frequency of LTG was slightly slower than that of LDG. LTG has several limitations in comparison to LDG, owing to technical difficulties in adequate LND and safe reconstruction. A longer learning curve might be needed to become an expert in LTG in comparison to LDG [25].

Gastric cancer treatment guidelines recommend harvesting at least 16 LNs for operable gastric cancer because inadequate LND can compromise oncological safety [17,18]. In our study, we found no significant difference in the mean numbers of dissected LNs between the OTG and LTG groups (37 LNs and 38 LNs, respectively). These numbers exceeded the amount of LND recommended in the guidelines. Additionally, several retrospective studies examined short-term surgical outcomes and showed that the LTG group had lower EBL but longer operating time than the OTG group [26,27]. Similar to previous studies, we showed that EBL was significantly lower in the LTG group than in the OTG group. However, there was no difference in the operating time between the two groups. In addition, some retrospective studies reported that the LTG group had shorter postoperative HD [26], and some studies did not [27]. Our study showed shorter HD in the LTG group than in the OTG group, but it was not a statistically significant difference (7 vs. 8 days, *p* = 0.1396).

Some retrospective studies demonstrated that LTG was superior to OTG in terms of complications [28,29,30]. However, several other studies suggested that there was no significant difference in complications between OTG and LTG. [26,31,32]. In our study, the overall complications were not significantly different between the two groups. There was significantly more grade I complications in the OTG group than in the LTG group. Grade I complications include minor complications, such as wound problems, urologic problems, and fluid collection, which do not require specific treatment. The incidence of other severe complications, including anastomotic leakage, was not significantly different between the groups. Therefore, these results suggested that LTG was not inferior to OTG in terms of short-term surgical outcomes.

Two meta-analyses compared 5-year survival between laparoscopic gastrectomy (LG) and open gastrectomy (OG) for AGC and demonstrated no significant difference between the two groups [33,34]. Additionally, two prospective studies conducted in Korea (KLASS-03) and Japan (JCOG1401) have successfully proven the safety of short-term surgical outcomes of LTG in EGC [35,36]. However, these two studies were limited to EGC and single-arm phase II studies. Furthermore, a randomized controlled study from China (CLASS-02) demonstrated the safety of LTG in EGC [37]. However, similar to the previous two studies, this trial was limited to EGC, and long-term survival data are not yet available. In AGC, there are only a few retrospective studies, with no prospective studies proving the safety of LTG. Therefore, the efficacy of LTG has not yet been sufficiently demonstrated.

Our study results showed that LTG was not inferior to OTG in terms of short- and long-term outcomes, regardless of the pathologic stage. LTG might be an alternative approach to OTG in gastric cancer. Two large prospective studies of the KLASS-06 (NCT03385018) and CLASS-07 (NCT04710758) trials are ongoing, and the results from these studies will provide strong evidence of the efficacy of LTG.

Our study had some limitations. First, it was a single-center retrospective study. Even though the bias correction was conducted by PSM, the selection bias of operation type on each patient in real world practice was inevitable if prospective randomized studies were not performed. Second, while this study was conducted in a large-volume center in Korea, the number of patients who underwent TG was still not enough to offer strong evidence and recommendation. Furthermore, the number of gastric cancer patients in the advanced stage was insufficient after matching. Third, all statistical biases could not be overcome even after PSM adjustment. Further large volume multicenter study will be needed to reinforce the evidence of this study.

## 5. Conclusions

LTG and OTG showed comparable efficacies in gastric cancer patients in terms of short- and long-term surgical outcomes. This study suggests that LTG could be an alternative approach to OTG.

## Figures and Tables

**Figure 1 cancers-15-00076-f001:**
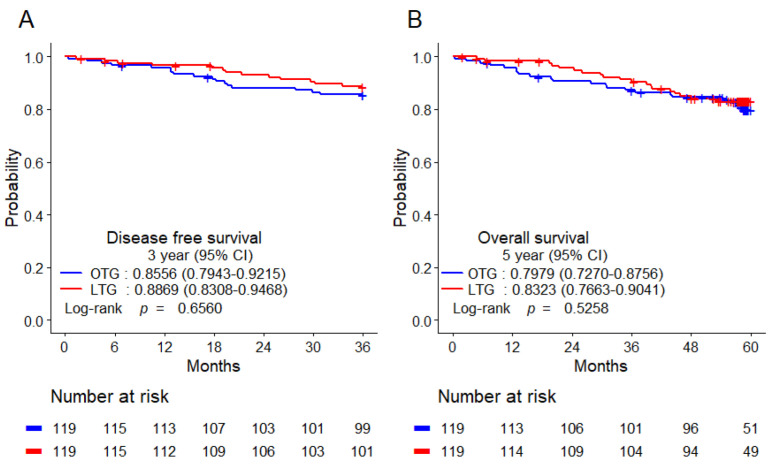
Kaplan−Meier survival analysis of 3-year DFS (**A**) and 5-year OS (**B**) after PSM. CI, confidence interval; DFS, disease-free survival; OS, overall survival; PSM, propensity score matching.

**Table 1 cancers-15-00076-t001:** Clinicopathological characteristics of patients.

Characteristic	Before PSM				After PSM			
	OTG (*n* = 409)	LTG (*n* = 151)	SMD	*p* Value	OTG (*n* = 119)	LTG (*n* = 119)	SMD	*p* Value
Age, years	59.5 ± 12.9	60.6 ± 12.0	0.085	0.3781	60.2 ± 11.6	62.3 ± 11.4	0.181	0.1645
Sex, *n* (%)			0.212	**0.0243**			0.020	0.8745
Male	295 (72.1)	94 (62.3)			94 (79.0)	93 (78.2)		
Female	114 (27.9)	57 (37.8)			25 (21.0)	26 (21.9)		
BMI	23.3 ± 3.5	23.4 ± 3.2	0.027	0.7841	23.8 ± 3.3	23.3 ± 3.1	0.151	0.2447
ASA score			0.074	0.4191			0.035	0.7897
I, II	389 (95.1)	141 (93.4)			112 (94.1)	111 (93.3)		
III	20 (4.9)	10 (6.6)			7 (5.9)	8 (6.7)		
Location			0.506	**0.0003**			0.497	**0.0032**
EG junction	91 (22.3)	18 (11.9)			24 (20.2)	14 (11.8)		
Upper	241 (58.9)	114 (75.5)			72 (60.5)	91 (76.5)		
Middle	49 (12.0)	19 (12.6)			14 (11.8)	14 (11.8)		
Lower	12 (2.9)	0 (0)			8 (6.7)	0 (0)		
Duodenum	5 (1.2)	0 (0)			0 (0)	0 (0)		
Whole stomach	11 (2.7)	0 (0)			1 (0.8)	0 (0)		
Histology			0.098	0.5169			0.198	0.2440
WD	72 (17.6)	32 (21.2)			29 (24.4)	30 (25.2)		
MD	106 (25.9)	38 (25.2)			38 (31.9)	32 (26.9)		
PD	135 (33)	47 (31.1)			27 (22.7)	34 (28.6)		
SRC	83 (20.3)	34 (22.5)			20 (16.8)	23 (19.3)		
Others	13 (3.2)	0 (0)			5 (4.2)	0 (0)		
Pathologic stage			1.541	**<0.0001**			**<0.001**	1
I	95 (23.2)	125 (82.8)			93 (78.2)	93 (78.2)		
II	119 (29.1)	17 (11.3)			17 (14.3)	17 (14.3)		
III	148 (36.2)	9 (6.0)			9 (7.6)	9 (7.6)		
IV	47 (11.5)	0 (0)			0(0)	0 (0)		
Tumor size			0.577	**<0.0001**			0.040	0.7569
< 5 cm	218 (53.3)	120 (79.5)			93 (78.2)	91 (76.5)		
≥ 5 cm	191 (46.7)	31 (20.5)			26 (21.9)	28 (23.5)		
LN Dissection			0.757	**<0.0001**			0.399	**0.0025**
< D2	49 (12)	66 (43.7)			29 (24.4)	51 (42.9)		
≥ D2	360 (88)	85 (56.3)			90 (75.6)	68 (57.1)		
Chemotherapy			1.371	**<0.0001**			0.126	0.3324
No	124 (30.3)	130 (86.1)			92 (77.3)	98 (82.4)		
Yes	285 (69.7)	21 (13.9)			27 (22.7)	21 (17.7)		

ASA, American Society of Anesthesiologists; BMI, body mass index; EG, esophagogastric; LTG, laparoscopic total gastrectomy; MD, moderately differentiated; OTG, open total gastrectomy; PD, poorly differentiated; PSM, propensity score matching; SMD, standard mean difference; SRC, signet ring cell carcinoma; WD, well differentiated. Bold values: *p* < 0.05.

**Table 2 cancers-15-00076-t002:** Postoperative surgical results.

Characteristic	Before PSM				After PSM			
	OTG (*n* = 409)	LTG (*n* = 151)	SMD	*p* Value	OTG (*n* = 119)	LTG (*n* = 119)	SMD	*p* Value
Operating time (min)	205 (85–605)	201 (115–480)	0.098	0.5169	200 (85–395)	205 (115–480)	0.198	0.2440
EBL (mL)	200 (0–2100)	50 (0–1050)	0.707	**<0.0001**	180 (0–1100)	50 (0–1050)	0.626	**<0.0001**
Positive LN	1 (0–117)	0 (0–20)	0.669	**<0.0001**	0 (0–26)	0 (0–20)	0.017	0.5135
Dissected LN	44 (0–154)	39 (0–97)	0.218	**0.0163**	37 (0–91)	38 (2–97)	0.088	0.3058
PRM (cm)	3 (0–18.5)	3.1 (0–16.5)	0.081	0.8352	3.7 (0–18.5)	3.1 (0–16.5)	0.368	**0.0491**
DRM (cm)	8.8 (0–27.8)	12.2 (0–25.7)	0.612	**<0.0001**	10 (0–27.8)	12.6 (0–25.7)	0.402	**0.0034**
HD (day)	8 (5–120)	7 (4–95)	0.050	**0.0005**	8 (5–120)	7 (4–95)	0.048	0.1396

DRM, distal resection margin; EBL, estimated blood loss; HD, hospital-stay duration; LN, lymph node; LTG, laparoscopic total gastrectomy; OTG, open total gastrectomy; PRM, proximal resection margin; PSM, propensity score matching; SMD, standard mean difference. Bold values: *p* < 0.05.

**Table 3 cancers-15-00076-t003:** Postoperative surgical complications after PSM.

Characteristics	OTG (*n* = 119)	LTG (*n* = 119)	*p* Value
Postoperative complication			
Wound infection	2(1.7)	1(0.8)	1
Fluid collection	7(5.9)	1(0.8)	0.0657
Inflammatory fluid collection	5(4.2)	0(0)	0.0599
Intraabdominal bleeding	1(0.8)	2(1.7)	1
Intestinal obstruction	3(2.5)	2(1.7)	1
Paralytic ileus	3(2.5)	3(2.5)	1
Anastomosis stenosis	1(0.8)	2(1.7)	1
Anastomosis leakage	3(2.5)	8(6.7)	0.1227
Atelectasis	1(0.8)	0(0)	1
Pneumonia	3(2.5)	5(4.2)	0.7219
Urologic	1(0.8)	0(0)	1
Hepatic	1(0.8)	0(0)	1
Cardiac	1(0.8)	0(0)	1
Pancreas	2(1.7)	0(0)	0.4979
Others	6(5.0)	5(4.2)	1
Duodenal stump leakage	1(0.8)	1(0.8)	1
Overall complication (%)	41(34.5)	30(25.2)	0.1191
Clavien–Dindo classification			
Grade I	13(11.0)	4(3.4)	**0.0224**
Grade II	8(6.8)	9(7.6)	0.8152
Grade IIIa	15(12.7)	9(7.6)	0.1889
Grade IIIb	3(2.5)	3(2.5)	1
Grade IV	1(0.9)	5(4.2)	0.2128
Grade V	1(0.9)	0(0)	0.4979
Clavien–Dindo classification ≥ IIIa	20(17.0)	17(14.3)	0.5722

LTG, laparoscopic total gastrectomy; OTG, open total gastrectomy; PSM, propensity score matching. Bold values: *p* < 0.05.

## Data Availability

The data presented in this study are available in this manuscript.

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
