# Peer review of "Short and Long-Term Surgical Outcomes of Laparoscopic Total Gastrectomy Compared with Open Total Gastrectomy in Gastric Cancer Patients"

_cancers, 2022, doi:10.3390/cancers15010076_

Round 1
Reviewer 1 Report
The authors tried to compare the outcomes of laparoscopic total gastrectomy (LTG) and open total gastrectomy (OTG) in gastric cancer patients, and they provided detailed data involving 560 patients before propensity-score matching (PSM) and 238 patients after PSM. The conclusion was concise and explicit. The main innovations of this work included two aspects, one was comparing the long-term outcomes of LTG and OTG, another was comparing the outcomes of LTG and OTG in advanced gastric patients. But the included patients in each group were only 9 after PSM, and more than 80% patients were stage I and stage II, which much impaired the reliability of the conclusions. It would be predictable that the rates of 3-year disease-free survival and 5-year overall survival in each group were both more than 80% and no statistical significance existed between LTG and OTG groups.
Author Response
Thanks for your kindness comments. I also agree with you.
As I was planning this article and collecting data, I was aware that the number of more advanced gastric cancers was too small after PSM
Since total gastrectomy has recently started to be performed in AGC in Korea, the number of patients was limited in order to check 5-year OS.
Therefore, there were limitations to evaluate the data.
Although there are differences in numbers between EGC and AGC, we tried to perform balanced matching in all gastric cancer stages including stage II.
Therefore, this article does not confirm, but suggests the efficacy of LTG.
Recently, many laparoscopic surgeries have been performed in AGC, we will try additional research in the future.
These contents were additionally described in disccussion.
Thank you.
Reviewer 2 Report
thank you for this review comparing laparoscopic to open total gastrectomy
1) In the abstract you refer to distal gastrectomy. Was it distal gastrectomy or total gastrectomy? Please correct this in the abstract as I think you meant to put total gastrectomy lap vs. open otherwise it would not be a fair comparison as these are very different procedures
2) In line 70 you state that open conversion was done for the Open group. what does that mean. Are you saying the lap cases that were converted were included in the open group? Please clarify
3) Is it possible that the less blood loss in the laparoscopic group was because there were fewer D2 dissections. As much as matching was attempted the groups were not perfectly matched.
Otherwise very interesting study. It is a shame that even short term outcomes were not found to be better with LSC surgery like in other cancers such as laparoscopic colon resection. No shorter length of stay or less pain. Did you look at pain in the short term? This should be commented on in the discussion.
Author Response
Thanks for your kindness comments
I agree with your comments
1) In the abstract you refer to distal gastrectomy. Was it distal gastrectomy or total gastrectomy? Please correct this in the abstract as I think you meant to put total gastrectomy lap vs. open otherwise it would not be a fair comparison as these are very different procedures
-> It seems that there was an error in the editing process. Sorry, I corrected it.
2) In line 70 you state that open conversion was done for the Open group. what does that mean. Are you saying the lap cases that were converted were included in the open group? Please clarify
-> It meant that if laparoscopic surgery was initially attempted, but when it was converted to open surgery, it was included in OTG group. I corrected the article.
3) Is it possible that the less blood loss in the laparoscopic group was because there were fewer D2 dissections. As much as matching was attempted the groups were not perfectly matched.
-> Estimated blood loss may be affected by lymph node dissection extent. However, this study was desinged to check the short and long-term outcomes results after mathcing the gastric cancer stage equally. So, it was excluded from matching variable.
Otherwise very interesting study. It is a shame that even short term outcomes were not found to be better with LSC surgery like in other cancers such as laparoscopic colon resection. No shorter length of stay or less pain. Did you look at pain in the short term? This should be commented on in the discussion.
-> It would have been nice to claim the advantages of laparoscopic surgery, but unfortunately, data about pain was not collected. And, hospital day was not significantly different between two groups. I additionally described it on discussion.
Round 2
Reviewer 1 Report
No more modifications needed.
Author Response
Thank you for your kind comment